# DNA vaccine candidate encoding SARS-CoV-2 spike proteins elicited potent humoral and Th1 cell-mediated immune responses in mice

Eakachai Prompetchara[1,2,3], Chutitorn Ketloy[1,2,3]*, Kittipan Tharakhet[1,2], Papatsara Kaewpang[1], Supranee Buranapraditkun[1,4], Teerasit Techawiwattanaboon[1,5], Suwitra Sathean-anan-kun[1,5], Patrawadee Pitakpolrat[1,2], Supaporn Watcharaplueksadee[6], Supaporn Phumiamorn[7], Wassana Wijagkanalan[8], Kanitha Patarakul[1,3,5], Tanapat Palaga[3,9], Kiat Ruxrungtham[1,3]

**1** Center of Excellence in Vaccine Research and Development (Chula Vaccine Research Center, Chula VRC), Faculty of Medicine, Chulalongkorn University, Bangkok, Thailand, **2** Department of Laboratory Medicine, Faculty of Medicine, Chulalongkorn University, Bangkok, Thailand, **3** Integrated Frontier Biotechnology for Emerging Disease, Chulalongkorn University, Bangkok, Thailand, **4** Department of Medicine, Faculty of Medicine, Chulalongkorn University, Bangkok, Thailand, **5** Department of Microbiology, Faculty of Medicine, Chulalongkorn University, Bangkok, Thailand, **6** Thai Red Cross Emerging Infectious Diseases-Health Science Centre, World Health Organization Collaborating Centre for Research and Training on Viral Zoonoses, King Chulalongkorn Memorial Hospital, Faculty of Medicine, Chulalongkorn University, Bangkok, Thailand, **7** Institute of Biological Product, Department of Medical Sciences, Ministry of Public Health, Nonthaburi, Thailand, **8** BioNet-Asia Co., Ltd, Bangkok, Thailand, **9** Department of Microbiology, Faculty of Science, Chulalongkorn University, Bangkok, Thailand

* Chutitorn.K@chula.ac.th

**Data Availability Statement:** All relevant data are within the paper and its Supporting Information file.

## Abstract

More than 65 million people have been confirmed infection with SARS-CoV-2 and more than 1 million have died from COVID-19 and this pandemic remains critical worldwide. Effective vaccines are one of the most important strategies to limit the pandemic. Here, we report a construction strategy of DNA vaccine candidates expressing full length wild type SARS-CoV-2 spike (S) protein, S1 or S2 region and their immunogenicity in mice. All DNA vaccine constructs of pCMVkan-S, -S1 and -S2 induced high levels of specific binding IgG that showed a balance of IgG1/IgG2a response. However, only the sera from mice vaccinated with pCMKkan-S or -S1 DNA vaccines could inhibit viral RBD and ACE2 interaction. The highest neutralizing antibody (NAb) titer was found in pCMVkan-S group, followed by -S1, while -S2 showed the lowest PRNT50 titers. The geometric mean titers (GMTs) were 2,551, 1,005 and 291 for pCMVkan-S, -S1 and -S2, respectively. pCMVkan-S construct vaccine also induced the highest magnitude and breadth of T cells response. Analysis of IFN-γ positive cells after stimulation with SARS-CoV-2 spike peptide pools were 2,991, 1,376 and 1,885 SFC/$10^6$ splenocytes for pCMVkan-S, -S1 and -S2, respectively. Our findings highlighted that full-length S antigen is more potent than the truncated spike (S1 or S2) in inducing of neutralizing antibody and robust T cell responses.

**Funding:** This study was funded by National Vaccine Institute (NVI), grant no. 2563. 1/ 8, National Research Council of Thailand (NTCT), Emerging Infectious Diseases and Vaccines Cluster, Ratchadapisek Sompoch Endowment Fund, Chulalongkorn University and Ratchadapiseksompotch Fund, Faculty of Medicine, Chulalongkorn University, grant no RA-MF-29/63. The authors would like to thank the Second Century Fund (C2F), Chulalongkorn University. EP was also supported by the Grants for Development of New Faculty Staff, Ratchadapiseksompote Endowment Fund, grant no DNS 63_031_30_009_2. WW is BioNet-Asia Co., Ltd consultant. The funder provided support in the form of budget for research materials but did not have any additional role in the study design, data collection and analysis, decision to publish, or preparation of the manuscript. The specific roles of the authors are articulated in the author contributions section.

**Competing interests:** The authors have read the journal's policy, and the authors of the study have the following competing interests to declare: WW is a consultant for BioNet-Asia Co., Ltd. BioNet-Asia Co., Ltd provided peptides for T cells assay. This does not alter our adherence to PLOS ONE policies on sharing data and materials. There are no patents, products in development or marketed products associated with this research to declare.

## Introduction

In December 2019, the outbreak of pneumonia caused by an unknown pathogen was documented in the city of Wuhan in China. The novel coronavirus, SARS-CoV-2, was then isolated and identified as a causative agent of the symptoms or disease, which was later named coronavirus disease 2019 or COVID-19 [1]. Due to the high volume of travel combined with advanced public transport, the disease spread globally in a short period of time. On March 11, 2020 it was declared a global pandemic [2, 3].

Currently more than 65 million confirmed cases which over 1.5 million deaths reported [4]. COVID-19 pandemic has thus led to major health, social and global economic crises. Vaccine is considered to be an effective strategy to control the pandemic. The vaccine for the current outbreak must be produced at a high speed and be easily scalable [5]. Recombinant protein, viral-vectored or nucleic acid-based (recombinant plasmid DNA and mRNA) vaccines should be considered to shorten the vaccine production timeframe [6, 7]. The lessons learnt and information from previous related coronaviruses outbreak, such as SARS in 2003 and MERS in 2012, revealed that the immune responses against viral spike (S) protein played an important role in viral infection inhibition [8–10]. Immunization of S protein induced potent neutralizing antibody (NAb) production and protected animals from viral challenge. The S protein is a trimeric transmembrane protein required for binding to its host receptor, angiotensin-converting enzyme 2 (ACE2), via the receptor-binding domain (RBD) located in the S1 region while the S2 region is responsible for virus-membrane fusion [10, 11]. Recently, almost 50 vaccine candidates are under clinical development [12, 13]. Phase I/II studies that used various versions of S protein antigen and production platform have been reported with satisfy safety and immunogenicity results [14–18]. There are at least 10 vaccine candidates testing in phase III trials [12, 19]. Any of these candidates may receive an emergency use approval for people at high risk by the end of 2020 or early 2021. Whether an effective vaccine against SARS-CoV-2 will be available sooner or later, scientific research to help further improvement of the next generation vaccine is warranted.

In the current study, we described the SARS-CoV-2 DNA vaccine candidate encoding different regions of SARS-CoV-2 S protein and evaluated their immunogenicity in mice. DNA vaccine platform was selected according to its advantages including rapid design and production and ability to induce both cellular and humoral immune responses [20]. More importantly, DNA vaccine avoid concern of Th2-skewed immune response which is the complication that have been observed in other respiratory viruses immunized with protein-based vaccines [21–23]. Synthetic DNA encoding different regions of S were subcloned into pCMVkan expression vector. We demonstrated *in vitro* target protein expression after transfection. The immunogenicity of the vaccine candidates was analyzed both for humoral and cell-mediated immune responses. Our data revealed the antibody responses including total IgG, IgG subclass as well as functional antibody (RBD-ACE2 binding inhibition and NAb). T cells response measured by IFN-γ secreted cells from mouse splenocytes after being stimulated with SARS-CoV-2 overlapping peptides was also demonstrated. The results provide insight information for antigen selection which could be generalized for further vaccine development during SARS-CoV-2 pandemic.

## Materials and methods

### Ethics statement

Animal experiments were performed in strict accordance with the recommendations of the Ethical Principles and Guidelines for the Use of Animals for Scientific Purposes. Mouse

experimental procedures and animal management in this study was approved by the Committee of Animal Care and Use, Faculty of Medicine, Chulalongkorn University (approval number 007/2563). Immunization and blood collections were performed under isoflurane anesthesia. All efforts to minimize the suffering of the animals were made throughout the study.

## Cell culture and viruses

The African Green Monkey Kidney (Vero, ATCC CCL81), HEK293T (ATCC® CRL-3216TM) were obtained from ATCC (Old Town Manassas, VA, USA). Vero and HEK293T cell lines were maintained in MEM and DMEM, respectively, and supplemented with 10% heat-inactivated fetal bovine serum (FBS), L-glutamine and penicillin-streptomycin (Gibco, USA). Cells were incubated at 37 $^O$C, 5% $CO_2$ atmosphere. The highly pathogenic SARS-CoV-2 virus was initially isolated from a clinical specimen of Thai patient-infected with SAR-CoV-2 virus, strain hCoV-19/Thailand/47/2020 by the National Institute of Health, Department of Medical Sciences, Thailand and was further propagated on Vero cells twice to obtain a large amount of viruses.

## Construction and preparation of recombinant plasmid DNA

To construct the recombinant plasmids, the protein sequence of SARS-CoV-2 S protein published in GenBank and GISAID during December 2019 to February 2020 were analyzed. Synthetic genes with humanized codon optimization encoding S, S1 or S2 were synthesized by GeneScript (Piscataway, NJ, USA) then subcloned into pCMVkan expression vector which contain the cytomegalovirus promoter, the bovine growth hormone polyadenylation site, and the kanamycin-resistant gene [24], designated as pCMVkan-S, pCMVkan-S1 and pCMVkan-S2, respectively, (nucleotide sequences are shown in S1 Data). In the S and S2 constructs, cytoplasmic region which contained ER-retention signal was deleted to enhance S protein trafficking to the cell surface. Signal peptide (SP), the first 15 amino acids at N-terminus of S protein, were included in all constructs (Fig 1A). The recombinant plasmids were propagated in E. coli, DH5-alpha (Invitrogen, Carlsbad, CA, USA) and purified by Qiagen endotoxin-free giga plasmid kit (Hilden, Germany) following the manufacturer's protocol. Characterization of the plasmids were performed by nucleotide sequencing and gel electrophoresis.

## Plasmids transfection and *in vitro* protein expression analysis

At 24 hr before transfection, $1x10^6$ cells of HEK293T were seeded in 6-well plate (Thermo Fisher Scientific, MA, USA). The cells with approximately 80–90% confluency were separately transfected with individual recombinant plasmid constructs (pCMVkan-S, -S1 and -S2) using lipofectamine 3000™ (Invitrogen, Carlsbad, CA, USA) according to the manufacture's protocol. Briefly, 2.5 μg of plasmid and 3.75 μL lipofectamine 3000™ were separately diluted in 125 μL and 250 μL serum-free Opti-MEM™ medium (Gibco), respectively. P3000™, a transfection enhancer reagent provided in lipofectamine 3000™ kit, was also added into diluted plasmid tube to a final concentration of 2 μL/μg DNA. The diluted DNA was mixed with the diluted lipofectamine 3000™ at 1:1 ratio (v/v) and incubated for 15 min at room temperature. The plasmid-lipofectamine 3000™ complex was then added onto cells. At 24 hr post-transfection, cells were fixed, permeabilized with ice-cold acetone and stained with 1:200 dilution of anti-S1 or anti-S2 polyclonal antibodies (Sino Biological, Beijing China). Donkey-anti-rabbit IgG-FITC, 1:5000 (BioLegend, USA), was used as a secondary antibody following anti-S1, or anti-S2 staining. Cell nuclei were counter stained with 4, 6-diamino-2-phenylindole hydrochloride (DAPI) (Sigma-Aldrich, USA). Stained cells were visualized under fluorescence microscope (Olympus, Japan).

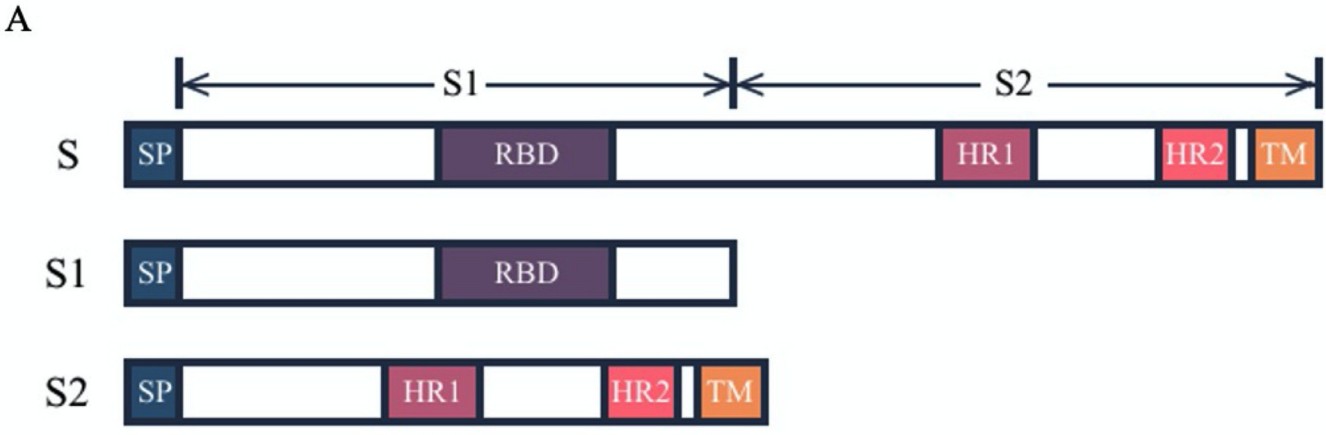

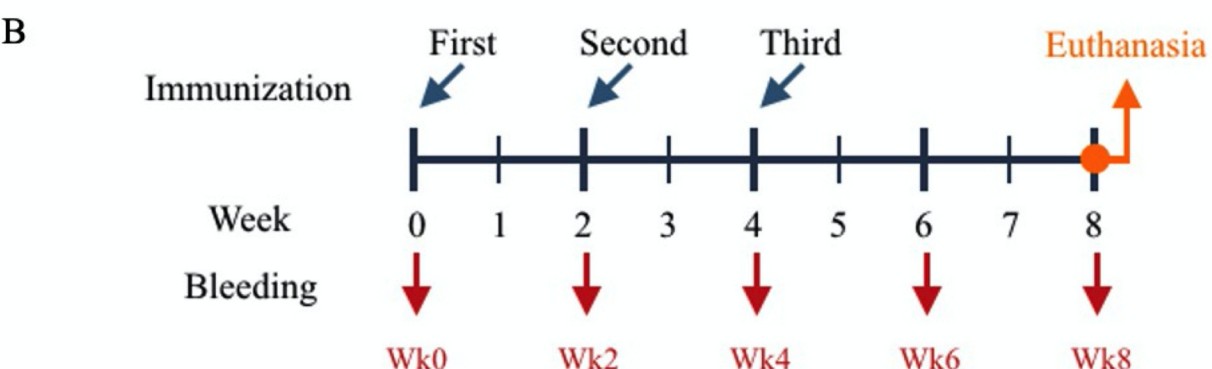

**Fig 1.** (A) Schematic diagram of three DNA vaccine constructs; SP: signal peptide, RBD: receptor binding domain, HR: heptad repeat, TM: transmembrane domain. (B) Mice immunization and sample collection schedule.

### Immunization of mice

Six-week old of female ICR mice (20–25 grams) were procured from the National Laboratory Animal Center, Mahidol University. Mice were randomly allocated into 4 groups with 5 mice/ group. Mice in each group were immunized with 100 μg of recombinant plasmids via intra-muscular electroporation using TriGrid delivery system (Ichor Medical System, CA, USA) (25). For negative control, 53.5 μg of empty pCMVkan which were equal molar with pCMVkan-S were used. The plasmids were administered 3 times, 2-week interval [25, 26]. Blood samples were collected at the day before each immunization, and 2 and 4 weeks after the $3^{rd}$ dose designated as weeks 0, 2, 4, 6 and 8. At week 8, the mice were euthanized by 30% $CO_2$ inhalation and splenocytes collected for T cells response analysis (Fig 1B).

### Enzyme-linked immunosorbent assay

End-point titers of total IgG as well as IgG1 and IgG2a isotypes in sera from immunized mice were measured by ELISA. Briefly, MaxiSorp™ flat-bottom 96-well plates (Nunc, Denmark) were coated with the mixture of SARS-CoV-2 S1 (RayBiotech, USA) and S2 proteins. The S2 protein was prepared by induction of the plasmid pET15b encoding S2 region fused with 6HIS

at N terminal. In brief, transformed *E. coli* BL21(*DE3*) were grown at 37˚C in LB (Luria-Bertani) medium supplemented with 100 μg/mL ampicillin. When the $OD_{600}$ of the cell culture reached 0.6, isopropyl β-D-1-thiogalactopyranoside, IPTG (final concentration 0.1 mM) was added to induce protein expression and cells were grown at 25˚C for a further 3 h. Cells were harvest by centrifugation at 8,000 g for 10 min at 4˚C. For S2 purification, the cell pellet was resuspended in PBS supplemented with DNase (10 μg/mL) and RNase (10 μg/mL). The cells were lysed by sonication on ice for 15 min. The lysed cells were then centrifuged at 40,000 rpm for 1 h, only supernatant was collected. S2 protein was purified by affinity chromatography using HisTrap column (GE Healthcare, Sweden). The concentration of purified protein was measured by using Pierce™ BCA protein assay kit (Thermo Fisher Scientific, MA, USA). In each well, the mixture of S1 (110 ng) and S2 (85 ng) which correspond to approximately 1.5 picomolar of each protein was diluted in 100 μL of coating buffer (0.1 M sodium carbonate) pH 9.5 and incubated overnight at 4˚C. Plates were then washed 5 times with PBS containing 0.05% Tween-20 (PBS-T), followed by blocking with 1% bovine serum albumin (BSA) in PBS-T for 1 h at 37˚C. After washing, plates were incubated with a 4-fold serial dilution of mouse sera (25 μL) starting from 1:100 to 1:102,400 and incubated for 1 h at 37˚C. The plates were then washed and incubated with 1:5000 dilutions of each horseradish peroxidase (HRP)-conjugated secondary Abs including rabbit anti-mouse IgG (KPL, USA), rabbit anti-mouse IgG1 or rabbit anti-mouse IgG2a (both Abs were from BioLegend, USA) for additional 1 h at 37˚C. After washing, 100 μL of tetramethylbenzidine (TMB) substrate (BioLegend, USA) was added and incubated for 5 min. The reactions were then stopped with 50 μL of 0.16 M sulfuric acid. The absorbance was measured by spectrophotometer at 450 nm using Varioskan microplate reader (ThermoFisher Scientific, Finland). End-point titers were determined and expressed as the reciprocals of the final dilution that emitted an optical density exceeding four-time of the background (BSA plus secondary antibody).

## RBD-ACE2 binding inhibition

ACE2 binding inhibition or surrogate viral neutralization test (sVNT) was performed according to the manufacture instruction (GenScript, Piscataway, NJ, USA). Briefly, diluted mice sera (1:100) were pre-incubated with RBD conjugated with HRP at 37 ˚C for 30 minutes. The mixtures were then put into hACE2 precoated plate and incubate at 37 ˚C for 15 minutes. After washed, 100 μL of TMB were added to each well and incubated at RT for 15 minutes in dark place. The reaction was stop by adding 50 μL of stop solution (0.2 N sulfuric acid), then read the absorbance at 450 nm immediately. Percent inhibition were calculated by comparing the OD value of sample and negative control.

## Plaque Reduction Neutralization Test (PRNT)

Vero cells were seeded at $6x10^5$ cells/3 ml in 6-well plates and incubated for 1 day at 37 ˚C with 5% $CO_2$. Test serum were initially diluted at 1:100 in the first wells followed by 4-fold serial dilutions up to 1:6400. SARS-CoV-2 virus was diluted in culture media to yield 40–120 plaques/well in the virus control wells. Cell control wells, convalescent patient serum and normal human serum were also included as assay controls. An equal volume of diluted SARS-CoV-2 virus was added to each diluted serum sample and the virus-serum mixture was incubated at 37 ˚C with 5% $CO_2$ for 1 h. The cell culture medium was then aspirated from the 6-well plates and the virus-serum mixture (200 ul/well) was added onto the plates and incubated at 37 ˚C with 5% $CO_2$ for 1 h for virus adsorption onto Vero cells. The virus-serum inoculum was then aspirated and 3 ml of 1% carboxymethylcellulose (CMC) overlay medium in MEM containing 1% L-glutamine and 5% FBS. The plates were incubated at 37 ˚C with 5%

$CO_2$ for 7 days. Cells were fixed with 10% (v/v) formaldehyde then stained with 0.5% crystal violet in PBS. The number of plaques was counted to determine the $PRNT_{50}$ titer. The neutralization titer ($PRNT_{50}$) of the test sample is defined as the reciprocal of the highest test serum dilution for which the virus infectivity is reduced by 50% when compared with the average plaque count of the virus control and was calculated by using a four-point linear regression method. Plaque counts for all serial dilutions of serum were scored to ensure that there was a dose response.

## Mouse IFN-γ ELISPOT assay

The SARS CoV-2 spike-specific IFN-γ secreting cells were determined by using a mouse IFN-γ ELISpot assay (Mabtech, Sweden). Briefly, splenocytes were resuspended at $5x10^6$ cells/ml in R5 medium. Ninety-six-well nitrocellulose membrane plates (MAHA S45; Millipore, USA) were coated with 50 μl/well of 10 μg/mL of anti-mouse IFN-γ (AN18) monoclonal antibody (mAb) (Mabtech, Sweden) in PBS at 37˚C with 5% $CO_2$ for 3 h. Then, the plates were washed six times with 200 μl PBS/well and blocked with 200 μL R10 medium/well at least 1 h at room temperature (RT). A quantity of $5x10^5$ splenocytes/well were cultured with SARS CoV-2 spike peptide pools (Mimotopes, Australia) at a final concentration of 2 μg/mL at 37 $^o$C with 5% $CO_2$ for 40 h, 10 peptide pools (containing 25–26 peptides/pool were used). The peptides were 15 amino acids (aa) overlapping by 10 amino acids spanning entire sequence of SARS-CoV-2 spike protein (n = 253). Culture medium alone and concanavalin A (ConA) served as negative and positive control, respectively. After incubation, the plates were washed then incubated with 1 μg/mL of anti-mouse IFN-γ-biotinylated mAb (R4-6A2 biotin; Mabtech, Stockholm, Sweden) in PBS for 3 h, RT. After washed, the plates were incubated with 50 μl/well of streptavidin-alkaline phosphatase (Mabtech, Sweden) diluted 1:1000 in PBS for 1 h, RT. The plates were washed, then 100 μL of the substrate solution (5-bromo-4-chloro-3-indolyl-phosphate/nitro blue tetrazolium; BCIP/NBT) were added into each well. The reaction was stopped by washing extensively in tap water and rinse the underside of membrane. Plates were then left dry and the spots counted using ELISpot reader (ImmunoSpot® Analyzer, Germany). Results were expressed as spot-forming cells (SFCs)/$10^6$ PBMCs after subtracted the spots from negative control wells.

## Cytokines measurement from splenocytes culture medium

Cytokines secretion collected from mice splenocytes grown in culture media was analyzed after the splenocytes were stimulated with spike pooled peptides for 40 h using the mouse Th1/Th2 Cytokine Panel (Biolegend, USA). The assay procedure was performed according to manufacturer protocol. The signals of each cytokine were analyzed by flow cytometry (BD FACS-Calibur, Becton Dickinson, USA).

## Statistical analysis

Statistical analysis was performed by using GraphPad Prism version 8 software. Comparisons of the data between group were made using Mann-Whitney tests. All $p$ values $< 0.05$ were defined as statistical significance.

## Results

### SARS-CoV-2 DNA vaccine candidates encoded the target protein *in vitro*

At 24 h post-transfection, transfected HEK293T were analyzed for *in vitro* target protein expression. As expected, protein expression after transfected with pCMVkan-S and -S1, but

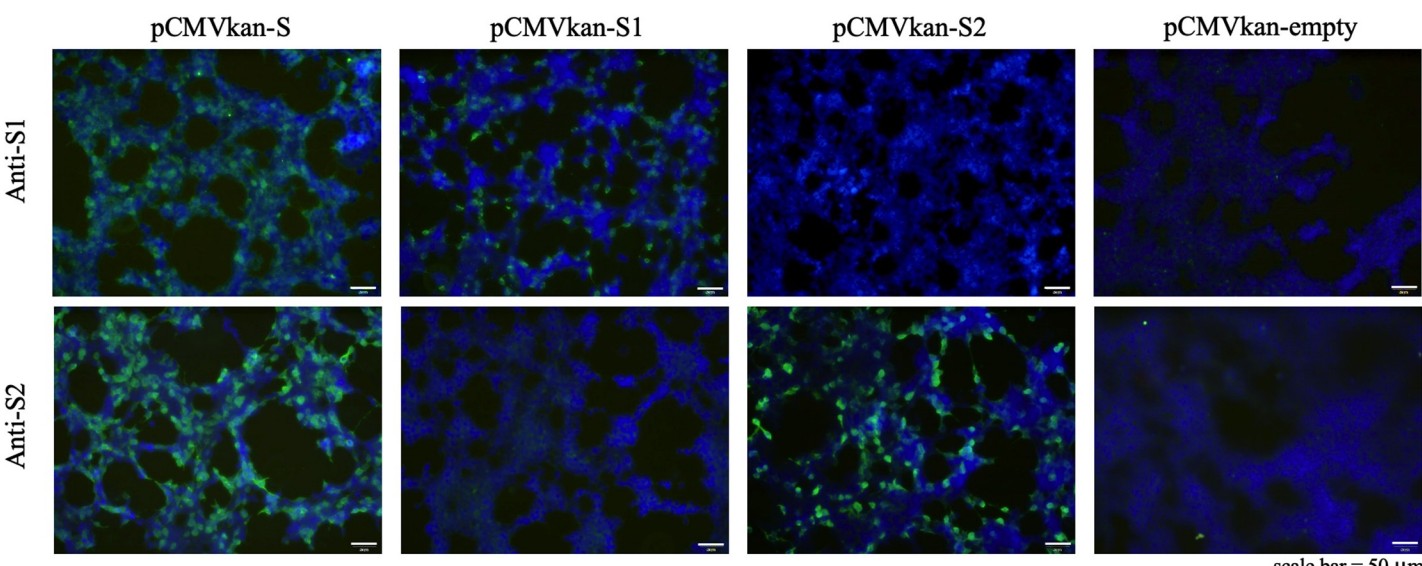

**Fig 2. Protein expression analysis in transfected cells.** HEK293T cells were transfected with pCMVkan-S, pCMVkan-S1, pCMVkan-S2 or empty vector. Target proteins expression were detected employing anti-S1(top) and anti-S2 (bottom) antibodies. Pictures were analyzed under fluorescent microscope with 10X objective lens.

not -S2 constructs could be detected by anti-S1 antibody. On the other hand, anti-S2 antibody could detect the target protein expression from cells transfected with pCMVkan-S and -S2, but not -S1 constructs. The percentage of S1 and S2 positive cells were comparable in pCMVkan-S transfected cells. There were no target protein expression from the negative control, empty pCMVkan transfected cells (Fig 2).

## S1/S2 protein-specific antibody responses were strongly elicited by SARS-CoV-2 DNA vaccine candidates

Mice sera were collected and analyzed serially before immunization and every two weeks after each vaccination. As shown in Fig 3, all vaccine constructs could augment S-specific antibody responses. The specific antibody titers could be detected after the first immunization was given and gradually increased following the later doses. S2 construct was found to be the most immunogenic candidate in terms of antibody response after administration of the 1st and 2nd doses. However, each vaccine construct showed comparable S-specific IgG titers after 3 doses (Fig 3A). Importantly, the analysis of IgG subclasses revealed comparable IgG1and IgG2a titers (Fig 3B).

## Antibodies induced by S and S1 DNA vaccine constructs could inhibit RBD-ACE2 binding

The interaction between viral RBD and ACE2 is considered as a surrogate SARS-CoV-2 neutralization event [27]. Hence, we analyzed the activity of immunized mice sera on the inhibition of RBD-ACE2 binding. The results demonstrated that mice that were immunized with S and S1 constructs could significantly inhibit RBD-ACE2 binding. At 1:100 dilution, the percent inhibition of the sera from mice immunized with S was significantly higher than those immunized with S1 (82% vs 56%) (Fig 4A). As expected, although the strong total specific IgG response was observed in S2 immunized sera, RBD-ACE2 binding inhibition could not be detected (Fig 4A).

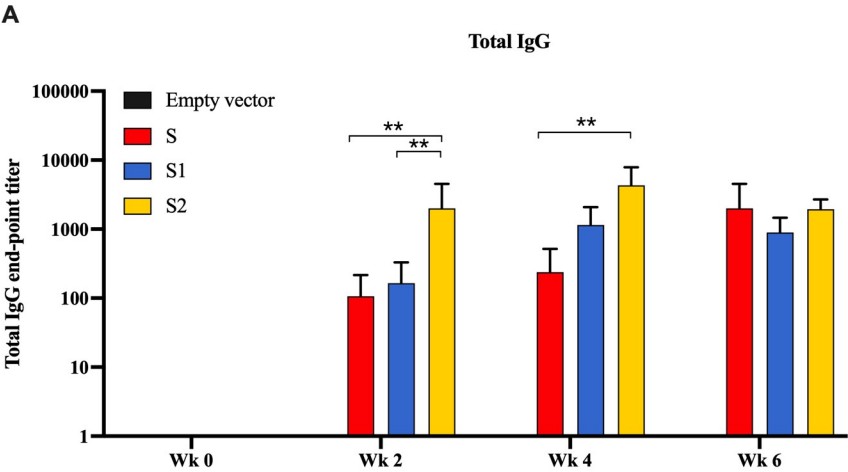

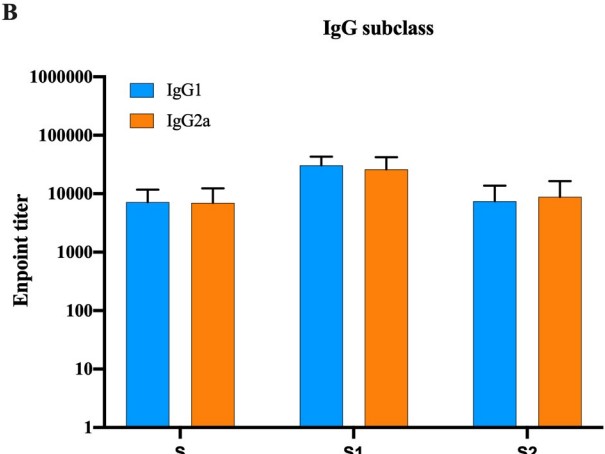

**Fig 3.** Titers of SARS-CoV-2 spike-specific total IgG analyzed at baseline and weeks 2, 4 and 6 (A), IgG subclass; IgG1 and IgG2a in the immunized mice sera collected on week 8 (B). Data presented as mean ± SD of the endpoint titers in each mice vaccination group (n = 5).

## SARS-CoV-2 DNA vaccine induced potent neutralizing antibody responses

Neutralizing antibody levels in immunized mice sera were analyzed by plaque reduction neutralizing assay (PRNT50) in Vero cells. At 2 weeks after the 3[rd] immunization, the most potent neutralizing activity was observed in mice immunized with S while S1immunized sera showed lower PRNT50 titers. The geometric mean titers (GMT) were 2,551 and 1,005 for S and S1 group, respectively (p<0.01). Consistent with RBD-ACE2 binding inhibition results, the neutralizing activity in S2 immunized mice was the lowest (GMT = 291) which is significantly lower than those of S and S1 immunized groups (Fig 4B).

## SARS-CoV-2 DNA vaccine candidates induced strong IFN-γ response

We measured IFN-γ secreting cells from splenocytes of immunized mice at 4 weeks after the last immunization using pooled peptides of S protein and ELISpot assay. The splenocytes from S-immunized mice secreted IFN-γ when stimulated with peptide pools both from S1 or S2

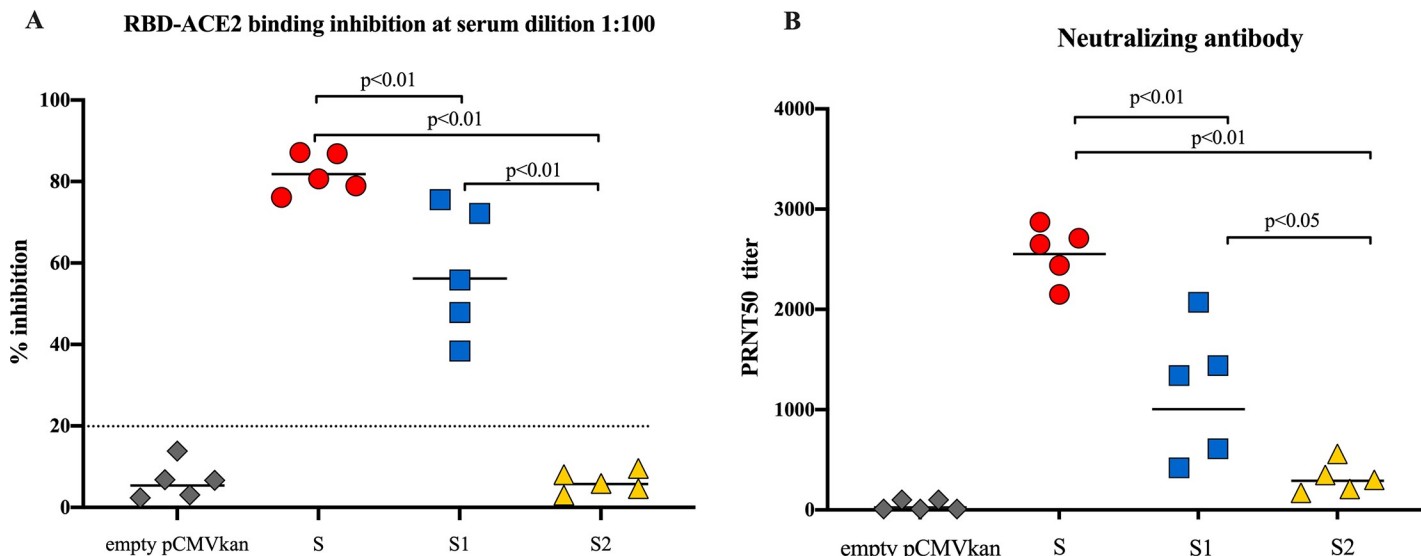

**Fig 4. Functional antibody analysis.** (A) Inhibition of RBD-ACE2 interaction; immunized mice sera collected at week 6 were diluted 100-fold and analyzed for their activity on the inhibition of RBD-ACE2 binding. Percent inhibition below the dash line is considered negative. (B) NAb responses measured by standard plaque assay. Data presented the PRNT50 titer of individual mice (n = 5), horizontal lines indicated the geometric mean.

regions. Mice that were immunized with S1 or S2 showed IFN-γ secreted positive cells only after stimulated with the corresponding peptide pools from each region. Higher magnitude of the IFN-γ ELISpot response was observed in the S-immunized mice (mean = 2,991 SCF/10$^6$ splenocytes) than those immunized with S1 (1,376 SCF/10$^6$ splenocytes) and S2 (1,885 SCF/10$^6$ splenocytes), p<0.05. IFN-γ ELISpot responses were strongest against peptide pool #3 corresponding to the RBD region and pool #8 corresponding to the HR1 region from S1 and S2, respectively, (Fig 5A–5C).

Besides the IFN-γ ELISpot assay, analysis of cytokines in supernatant secreted by splenocytes from the S-immunized group upon stimulation *ex vivo* with SARS-CoV-2 pooled peptides, showed high levels of IL-2 secretion but low levels of IL-4 (Fig 5D). This result indicates a favorable Th1 profile of T cell response after pCMVkan-S immunization.

## Discussion

There are almost 200 candidate vaccines under development based on 6 technology platforms: inactivated vaccines, protein subunit vaccines, mRNA vaccines, DNA vaccines, non-replicating/replicating viral vector vaccines, and viral-like particle vaccine, (WHO landscape report by November 3, 2020) (13). Among these, 47 candidates are entering clinical development (phase 1–3). The preliminary results of these vaccines showed satisfying immune responses in both preclinical and early clinical trial phases. For example, an mRNA vaccine developed by Moderna, expressing whole spike with 2-proline mutations (mRNA-1273) to stabilize the spike protein in prefusion form [28, 29], showed a robust NAb response and protection after viral challenge in mouse strains and non-human primates. CD4$^+$ T cell and T follicular helper cell responses were also detected [29]. Wild-type version of spike protein used in viral vector vaccines, such as ChAdOx1 nCoV-19, developed by the University of Oxford and AstraZeneca was also tested. Immunized macaques developed NAb and Th1 responses and were mostly protected from viral replication after challenged [30]. Other spike versions such as RBD trimerization domain using mRNA platform (BNT162b1) developed by BioNTech-Pfizer was also evaluated. Recently, Pfizer reported a direct comparison between BNT162b1 with

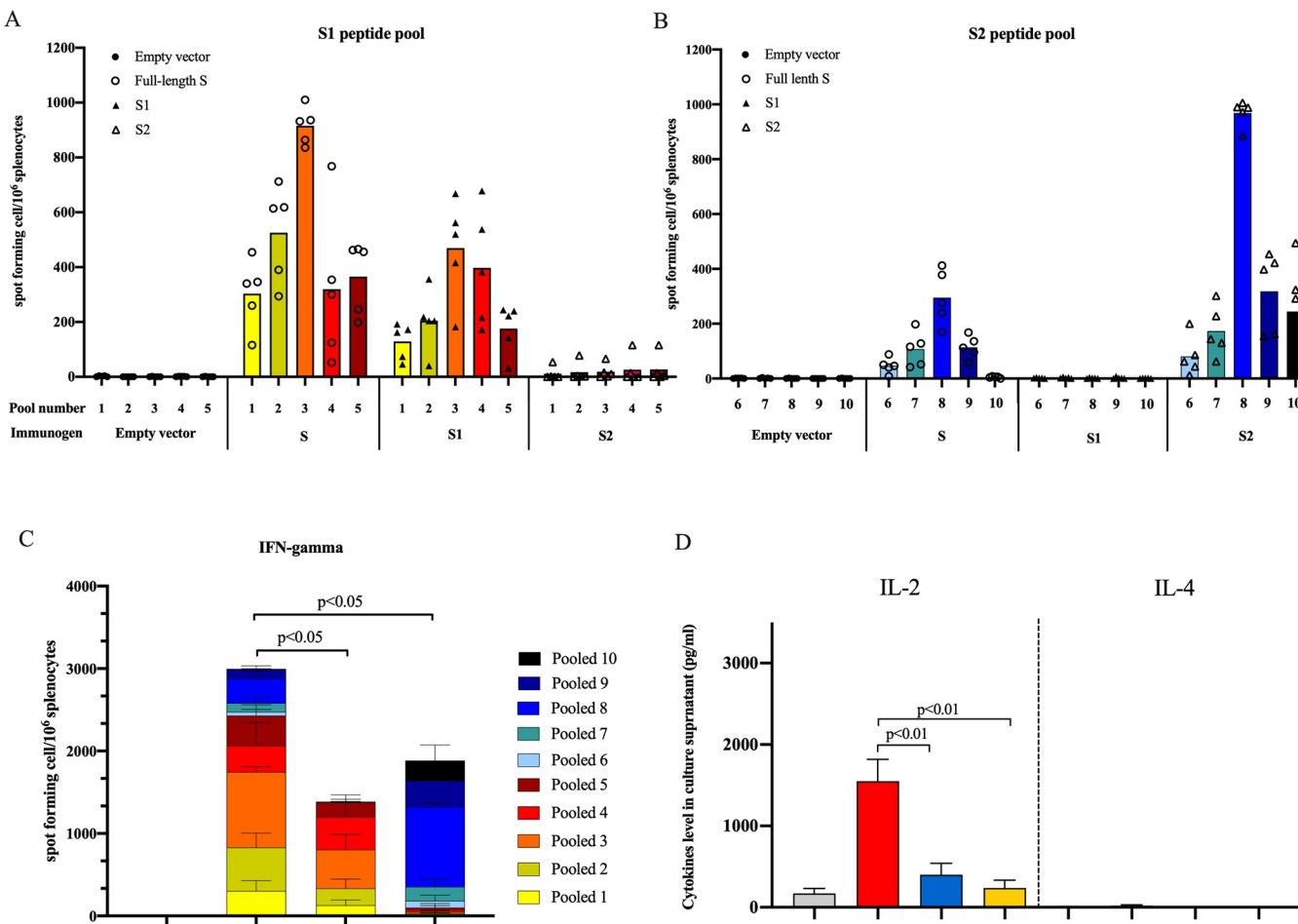

**Fig 5. SARS-CoV-2 spike-specific T cells responses analyzed by ELISpot.** Mice splenocytes were simulated with pooled peptides from S1 (A) or S2 (B) regions. Each bar represents mean of IFN- γ spot forming unit (SPF) per million splenocytes responses to each peptide pool. (C) Sum of IFN- γ responses in mice (N = 5) immunized with different DNA vaccine constructs. Mann-Whitney test was used to compare the different in the total IFN- γ responses from each group. (D) *Ex vivo* cytokines expression collected from the culture supernatant after splenocytes from S-immunized group were stimulated with spike pooled peptides for 24 h.

BNT162b2 which encodes a full-length spike protein with di-proline mutations. Results showed comparable antibody titers but BNT162b2 provided a more favourable safety profile [31]. Although the NAb titers were reported in all vaccine candidates, the results are difficult to compare due to different techniques and readouts in NAb measurement [32].

In this study, we selected DNA vaccine platform as its advantage is that it could be produced in a limited short time [33]. The plasmid backbone and expression system used in this was previously proof for its efficacy in dengue DNA vaccine [25, 34]. The objective was to investigate the immunogenicity of SARS-CoV-2 spike either as a full-length wild type or truncated as S1 or S2 in this DNA vaccine design platform in murine model.

Here, we demonstrated that SARS-CoV-2 DNA vaccine candidates elicited significant viral-specific humoral and cell-mediated immune responses. Although RBD is currently the main target antigen in multiple vaccine candidates, we selected S or S1 rather than RBD because it had previously been reported that the potent SARS-CoV-2 neutralizing epitopes also located outside the RBD region. For example, analysis of COVID-19 patients sera revealed

that neutralizing antibodies were also directed against epitopes in N-terminal [35] and C-terminal [36, 37] domain of S1 protein. Prior study also agreed with this finding that the recombinant S1 protein is more immunogenic than the RBD when total IgG, IgA and NAb were analyzed [38]. Analysis of IgG subclass in all vaccine constructs revealed a mixed response of IgG2a and IgG1(Fig 3B). This implied that our SARS-CoV-2 DNA vaccine candidates induced a balanced Th1/Th2 responses that could avoid the concern of vaccine-associated enhanced respiratory disease (VAERD) induced by Th2-bias vaccine modality [21].

Although the total spike protein binding antibody was found at a similar level in mice when S-, S1- or S2-DNA vaccine candidates were administered (Fig 3), the significantly highest live virus neutralization activity was found in the S-DNA vaccinated group. Lower neutralizing activity was probably due to the lack of trimeric structure of the S1 construct [11, 39]. On the other hand, S construct potentially forms a trimeric structure similar to those of the spike on the viral structure [11, 39]. Moreover, the superiority of S over S1 construct might also be mediated by the inclusion of S2 region [11, 40]. As shown in the result, S2 is highly immunogenic when analyzed by ELISA and ELISpot but low levels NAb was detected. Hence, antibody against S2, at least, might help to enhance viral neutralization activity. For example, antibody directed to fusion peptides could prevent virus-cell fusion [41]. Neutralizing activity of the antibody against fusion peptides was revealed from COVID-19 patient sera [42]. Moreover, antibody against the connecting heptad repeat (HR) 1 and HR2 (amino acid 1148–1159) on the S2 subunit also mediated viral neutralization [37].

Cell-mediated immune response also plays a critical role in viral control as the cellular immune response without antibody response was found in asymptomatic and mild symptoms COVID-19 diagnosis confirmed individuals [43]. Moreover, a previous study showed most of patients who recovered from COVID-19 had S protein-specific CD4+ and CD8+ T cells in the peripheral blood [44]. Nonetheless, its role in protection against SARS-CoV-2 infection remained to be elucidated. A previous study of a SARS-CoV-1 vaccine study in mice supported a protective role of T cells [45]. In this study we found that the highest magnitude of SARS-CoV-2-specific T cells responses was observed in wild-type S-spike DNA immunization. The most abundant responses were directed to RBD and HR1peptide pools. This finding is consistent with the results from a phase 1 clinical trial that participants who were immunized with a di-proline spike mRNA vaccine candidate generated good CD4+ T cells responses against both S1 and S2 pooled peptides [16]. Moreover, previous studies on T cells epitope mapping both *in vivo* and *in silico* revealed that the immunodominant T cells epitopes also found in RBD and HR1 regions [46, 47].

In our study, however, provided only overall splenocytes responses, detailed investigation on responses of different helper T cell subsets will provide more precise information on DNA vaccine-induced T cell immunity. The cytokine profile analysis in the supernatant of cultured splenocytes restimulated with SARS-CoV-2 overlapping peptides revealed the Th1-skewed response in this S-DNA vaccine candidate. Thus, this finding may minimize the concern the immunopathology caused by Th2-bias response previously found in MERS and SARS CoV-1 vaccines [48–50].

As reported in MERS and SARS CoV-1, antibody-dependent enhancement, ADE is one of the concerns in vaccine-induced antibody response [51–53]. However, recent studies demonstrated that unlike in MERS and SARS CoV-1, the full-length S or RBD of SARS CoV-2 elicited a robust neutralizing antibody response without inducing ADE in animal immunization studies [54, 55]. To confirm this concern, analysis of ADE phenomenon is warranted especially in clinical trial studies.

The full-length wild type spike used in this study had previously proved that it could form a trimeric structure identical to those of 2-proline mutations [56]. The immunogenicity as well

as protective efficacy of full-length wild type spike was also demonstrated in various animal models [30, 57, 58]. The study in phase I clinical trial demonstrated that DNA vaccine using full-length wild type spike was safe and could induce both antibodies and T cells responses [59]. ChAdOx1 nCoV-19, the most advanced vaccine candidate that used full-length wild type spike as an antigen is being tested in phase III clinical trials (ISRCTN89951424, NCT04516746). Recently, interim analysis of ChAdOx1 nCoV-19 vaccine trials showed that the overall efficacy of ChAdOx1 nCoV-19 was 70.4% [60]. This led to emergency use approval of ChAdOx1 nCoV-19 in Britain, India and other countries. Thus, these findings confirmed that the full-length wild type spike is also served as a good candidate antigen that could provide high level of protection in large clinical trials.

Due to the limitation of the ABSL-3 facility, a challenge study of these vaccine candidates could not be performed in this study. However, by comparing with the study that could convert the readout of NAb levels to PRNT titers, the PRNT titer at average of $3\log_{10}$ could protect mice from respiratory tract infection [61]. This implies that the NAb titers induced by pCMVkan-S DNA immunization in this study is potentially sufficient to provide protection. However, as the correlated protective immunity has not been established for COVID-19, a challenge study in animal model to provide efficacy data is still required.

In conclusion, this study provides crucial information regarding selection of antigens for SARS-CoV-2 vaccine development. SARS-CoV-2 full length S (S1+S2) is more potent in induction of both NAb and T cells responses than the truncated S1 or S2 immunogen. This finding could be further applied for development of other vaccine modalities.

## Supporting information

**S1 Data.**
(DOCX)

## Acknowledgments

We would like to thank Dr. Barbara K. Felber (the NCI-FCRDC, USA) for providing the pCMVkan expression vector, The Department of Disease Control, Ministry of Public Health, Thailand for providing clinical specimens for the viral isolate. We are also grateful Dr. Navapon Techakriengkrai for providing HEK293T cells.

## Author Contributions

**Conceptualization:** Eakachai Prompetchara, Chutitorn Ketloy, Wassana Wijagkanalan, Kanitha Patarakul, Tanapat Palaga, Kiat Ruxrungtham.

**Data curation:** Eakachai Prompetchara, Chutitorn Ketloy, Kittipan Tharakhet, Papatsara Kaewpang, Supranee Buranapraditkun, Teerasit Techawiwattanaboon.

**Formal analysis:** Eakachai Prompetchara, Chutitorn Ketloy, Supranee Buranapraditkun, Teerasit Techawiwattanaboon.

**Funding acquisition:** Eakachai Prompetchara, Chutitorn Ketloy, Kiat Ruxrungtham.

**Investigation:** Eakachai Prompetchara, Chutitorn Ketloy, Kittipan Tharakhet, Papatsara Kaewpang, Supranee Buranapraditkun, Teerasit Techawiwattanaboon, Suwitra Satheananan-kun, Patrawadee Pitakpolrat, Supaporn Watcharaplueksadee, Supaporn Phumiamorn.

**Methodology:** Eakachai Prompetchara, Kittipan Tharakhet, Papatsara Kaewpang, Supranee Buranapraditkun, Teerasit Techawiwattanaboon, Suwitra Sathean-anan-kun, Patrawadee Pitakpolrat, Supaporn Watcharaplueksadee, Supaporn Phumiamorn.

**Resources:** Wassana Wijagkanalan.

**Supervision:** Chutitorn Ketloy, Kanitha Patarakul, Tanapat Palaga, Kiat Ruxrungtham.

**Visualization:** Kittipan Tharakhet.

**Writing – original draft:** Eakachai Prompetchara.

**Writing – review & editing:** Eakachai Prompetchara, Chutitorn Ketloy, Kanitha Patarakul, Tanapat Palaga, Kiat Ruxrungtham.

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
