## [Decision Letter · Decision Letter 0]

7 Jan 2021

PONE-D-20-38402

DNA vaccine encoding SARS-CoV-2 spike proteins elicited potent humoral and Th1 cell-mediated immune responses in mice

PLOS ONE

Dear Dr. Ketloy,

Thank you for submitting your manuscript to PLOS ONE. After careful consideration, we feel that it has merit but does not fully meet PLOS ONE’s publication criteria as it currently stands. Therefore, we invite you to submit a revised version of the manuscript that addresses the points raised during the review process:

1) Please, consider changing the title as suggested by reviewer #2;

2) The authors should provide more methodological details in order to allow others to repeat and confirm the results;

3) Please, answer to all the comments raised by the both reviewers.

We look forward to receiving your revised manuscript.

Kind regards,

Paulo Lee Ho, Ph.D.

Academic Editor

PLOS ONE

Journal Requirements:

2.) Thank you for including your ethics statement:  "Animal experiments were performed in strict accordance with the recommendations of the Ethical Principles and Guidelines for the Use of Animals for Scientific Purposes. Mouse experimental procedures and animal management were undertaken in accordance with the requirement of the Committee of Animal Care and Use, Faculty of Medicine, Chulalongkorn University (approval number 007/2563). Immunization and blood collections were performed under isoflurane anesthesia. All efforts to minimize the suffering of the animals were made throughout the study.

Please amend your current ethics statement to confirm that your named ethics committee specifically approved this study.

For additional information about PLOS ONE submissions requirements for ethics oversight of animal work, please refer to http://journals.plos.org/plosone/s/submission-guidelines#loc-animal-research  

3.) In your Methods section, please provide methods of animal sacrifice and efforts to alleviate suffering.

4.) Thank you for stating the following in the Competing Interests section:

'This study was funded by National Vaccine Institute (NVI), grant no.2563.1/8, National Research Council of Thailand (NTCT), Ratchadapiseksompotch Fund, Faculty of Medicine, Chulalongkorn University, grant no RA-MF-29/63 and Second Century Fund (C2F), Chulalongkorn University.  EP was also supported by the Grants for Development of New Faculty Staff, Ratchadapiseksompote Endowment Fund, grant no DNS 63_031_30_009_2.'

We note that one or more of the authors are employed by a commercial company: BioNet-Asia Co., Ltd,

5.) PLOS requires an ORCID iD for the corresponding author in Editorial Manager on papers submitted after December 6th, 2016. Please ensure that you have an ORCID iD and that it is validated in Editorial Manager. To do this, go to ‘Update my Information’ (in the upper left-hand corner of the main menu), and click on the Fetch/Validate link next to the ORCID field. This will take you to the ORCID site and allow you to create a new iD or authenticate a pre-existing iD in Editorial Manager. Please see the following video for instructions on linking an ORCID iD to your Editorial Manager account: https://www.youtube.com/watch?v=_xcclfuvtxQ

Reviewers' comments:

Reviewer's Responses to Questions

**Comments to the Author**

1. Is the manuscript technically sound, and do the data support the conclusions?

Reviewer #1: Yes

Reviewer #2: Yes

2. Has the statistical analysis been performed appropriately and rigorously? 

Reviewer #1: Yes

Reviewer #2: Yes

3. Have the authors made all data underlying the findings in their manuscript fully available?

Reviewer #1: Yes

Reviewer #2: No

4. Is the manuscript presented in an intelligible fashion and written in standard English?

Reviewer #1: Yes

Reviewer #2: Yes

5. Review Comments to the Author

Reviewer #1: The present study report results obtained by DNA vaccination using 3 different proteins from SARS-COV19 Spike – S (whole protein), S1 and S2. Antibodies and cellular response were generated, showing better results for the whole protein construction. The manuscript is well written, the results are expected but they are not novelty, comparing with other vaccine studies using SARS-COV 19 spike protein.

DNA vaccines are easy to obtain and produce, are stable and stimulates both arms of immune response. It is well known. However, they still have technical problems that have not been solved until today – DNA vaccines working very well in murine models with results that are not repeated in primates, human and non-human. So far, we only have DNA vaccines approved for veterinary use.

With these considerations, DNA vaccines at the moment serve more as proof of principle, which in this case (SARS-COV19) is already well known with other vaccine models.

Other concerns

Material and methods are poorly described. Reagent quantities are not described in several protocols; necessary corrections must be made. A methodology needs to be well described in order to allow other researchers to be able to reproduce them.

Line 122 - Plasmids transfection and in vitro protein expression analysis - Protocols must be better described. Please, inform with more details the quantities of reagents were used in this assays. How much cells were transfected??? How much plasmid, lipofectamine and IgG-FITC were used???

Line 131

Please include the average weight of the animals.

Were the blood samples taken on the same day as the immunizations? Please clarify.

Line 140 - ELISA:

Please, inform antigen concentrations in micrograms/well used. How much of sera was used, 100 microliters? Please, specify.

Please inform quantity (titer) of HRP anti-mouse IgG was used.

How much of TMB was used???

Line 158 - RBD-ACE2 binding inhibition

How much of diluted mice sera (what dilution???) and RBD-HRP were used???

TMB??? What volume was used???

Line 282 : Please, describe what NHP means (non human primates???)

Reviewer #2: Prompetchara et al describe the production and the use of plasmids containing full-length, just the S1, or S2 portions of the S1-glycoprotein of SARS-Cov2 in generating immune responses in mice. Similar studies have shown the possibility to generate immune responses in mice, guinea pig and Rhesus monkey. Recently, one of them has been adapted to be used in humans (https://doi.org/10.1016/j.eclinm.2020.100689), which could be additionally cited by the authors.

The value of these manuscript could be to help in devising alternative ways to generate a protective response against the virus for human subjects and be used for vaccination.

The results show improved responses with the use of full length spike, which should be better highlighted in the manuscript.

Minor issues

I would replace the title for DNA plasmid as there is not in vivo challenging in this work. Otherwise, use vaccine candidate.

Line 38: define GMT in the abstract

Line 40: stimulation

Methods

Plasmid nucleotide sequence details or deposit in public data bank should be provided. It is remarkable that many works have not disclosed this information, which I fell incorrect in scientific publications.

The reference of pCMVkan should be cited in methods.

Details about transfections (amounts and purification of DNA plasmids) are important to be mentioned here.

In house preparation of S2 should be further explained.

Results

Fig. 2: size bars are poorly visible and

Fig. 3: It is not clear which assay was used for IgG1xIgG2, please explain.

6. PLOS authors have the option to publish the peer review history of their article (what does this mean?). If published, this will include your full peer review and any attached files.

Reviewer #1: No

Reviewer #2: No

---

## [Author Response · Author response to Decision Letter 0]

27 Jan 2021

Re- PONE-D-20-38402

Dear Editor,

Thank you very much for the useful comments from editor and reviewers. On behalf of the authors, I’d like to respond as following:

Editor comments:

1. Please, consider changing the title as suggested by reviewer #2;

Response: the title was revised to “DNA vaccine candidate encoding SARS-CoV-2 spike proteins elicited potent humoral and Th1 cell-mediated immune responses in mice”

2) The authors should provide more methodological details in order to allow others to repeat and confirm the results;

Response: See the responses on the methodological details to both Reviewer#1 and Reviewer#2 below

3) Please, answer to all the comments raised by the both reviewers.

Response: See the response to both Reviewer#1 and Reviewer#2 below

Reviewer#1

• Line 122 - Plasmids transfection and in vitro protein expression analysis - Protocols must be better described. Please, inform with more details the quantities of reagents were used in this assays. How much cells were transfected??? How much plasmid, lipofectamine and IgG-FITC were used???

Response: The details for plasmid transfection and in vitro protein expression was described.

“At 24 hr before transfection, 1x106 cells of HEK293T were seeded in 6-well plate (Thermo Fisher Scientific, MA, USA). The cells with approximately 80-90% confluency were separately transfected with individual recombinant plasmid constructs (pCMVkan-S, -S1 and -S2) using lipofectamine 3000™ (Invitrogen, Carlsbad, CA, USA) according to the manufacture’s protocol. Briefly, 2.5 µg of plasmid and 3.75 µL lipofectamine 3000™ were separately diluted in 125 µL and 250 µL serum-free Opti-MEM™ medium (Gibco), respectively. P3000™, a transfection enhancer reagent provided in lipofectamine 3000™ kit, was also added into diluted plasmid tube to a final concentration of 2 µL/µg DNA. The diluted DNA was mixed with the diluted lipofectamine 3000™ at 1:1 ratio (v/v) and incubated for 15 min at room temperature. The plasmid-lipofectamine 3000™ complex was then added onto cells.” (Line 136-146)

• Line 131 Please include the average weight of the animals.

Response: Mouse weight (20-25 grams) was included (Line 153).

• Were the blood samples taken on the same day as the immunizations? Please clarify.

Response: The detail for blood sample collection was clarified (Line 160-161).

“Blood samples were collected at the day before each immunization, and 2 and 4 weeks after the 3rd dose designated as weeks 0, 2, 4, 6 and 8.”

• Line 140 - ELISA:

Please, inform antigen concentrations in micrograms/well used. How much of sera was used, 100 microliters? Please, specify.

Please inform quantity (titer) of HRP anti-mouse IgG was used.

How much of TMB was used???

Response: The detail for ELISA protocol was provided in the method including 

- antigen concentrations in micrograms/well used 

“In each well, the mixture of S1 (110 ng) and S2 (85 ng) which correspond to approximately 1.5 picomolar of each protein was diluted in 100 µL of coating buffer” (Line 177-178)

- amount of sera 

“25 µL” (Line 182)

- quantity (titer) of HRP anti-mouse IgG 

“1:5000 dilutions of each horseradish peroxidase (HRP)-conjugated secondary Abs” (Line 183)

- Amount of TMB used 100 µL of tetramethylbenzidine (TMB) substrate (Line 186)

• Line 158 - RBD-ACE2 binding inhibition

How much of diluted mice sera (what dilution???) and RBD-HRP were used???

Response: The detail for RBD-ACE2 binding inhibition protocol was provided in the method including

- dilution of sera 

diluted mice sera (1:100) (Line 195)

- the use of RBD-HRP 

RBD conjugated with HRP (Line 195) 

- Volume of TMB used 

100 µL of TMB (Line 197)

- Volume of stop solution 

50 µL of stop solution (Line 198)

• Line 282 : Please, describe what NHP means (non human primates???)

Response: Yes, NHP is stand for non-human primates (Line 316)

Reviewer #2: 

• Prompetchara et al describe the production and the use of plasmids containing full-length, just the S1, or S2 portions of the S1-glycoprotein of SARS-Cov2 in generating immune responses in mice. Similar studies have shown the possibility to generate immune responses in mice, guinea pig and Rhesus monkey. Recently, one of them has been adapted to be used in humans (https://doi.org/10.1016/j.eclinm.2020.100689), which could be additionally cited by the authors.

Response: The recent study in phase I clinical is cited 

“The study in phase I clinical trial demonstrated that DNA vaccine using full-length wild type spike was safe and could induce both antibodies and T cells responses (59)” (Ref 59, Line 388-390)

• The results show improved responses with the use of full length spike, which should be better highlighted in the manuscript.

Response: We highlighted the use of wild-type full length which it could provide high level of protective efficacy as seen in ChAdOx1 nCoV-19 (in discussion Line 392-396). And based on our findings, in conclusion we also state that full length S (S1+S2) is better than the truncated S1 or S2 immunogens (Line 408-401).

• I would replace the title for DNA plasmid as there is not in vivo challenging in this work. Otherwise, use vaccine candidate.

Response: the title was revised to “DNA vaccine candidate encoding SARS-CoV-2 spike proteins elicited potent humoral and Th1 cell-mediated immune responses in mice”

• Line 38: define GMT in the abstract

Response: Geometric mean titers added (Line 49)

• Line 40: stimulation

Response: corrected to stimulation (Line 52)

• Plasmid nucleotide sequence details or deposit in public data bank should be provided. It is remarkable that many works have not disclosed this information, which I fell incorrect in scientific publications.

Response: As mentioned in method section, we used the sequence from online database which is the wild-type sequence without further modification of amino acids. So, the sequence still the same with the uploaded database.

• The reference of pCMVkan should be cited in methods.

Response: The information of pCMVkan and reference was included 

“pCMVkan expression vector which contain the cytomegalovirus promoter, the bovine growth hormone polyadenylation site, and the kanamycin-resistant gene (24)” (Ref 24, Line 125-127)

• Details about transfections (amounts and purification of DNA plasmids) are important to be mentioned here.

Response: The details for plasmid transfection and in vitro protein expression was described 

“At 24 hr before transfection, 1x106 cells of HEK293T were seeded in 6-well plate (Thermo Fisher Scientific, MA, USA). The cells with approximately 80-90% confluency were separately transfected with individual recombinant plasmid constructs (pCMVkan-S, -S1 and -S2) using lipofectamine 3000™ (Invitrogen, Carlsbad, CA, USA) according to the manufacture’s protocol. Briefly, 2.5 µg of plasmid and 3.75 µL lipofectamine 3000™ were separately diluted in 125 µL and 250 µL serum-free Opti-MEM™ medium (Gibco), respectively. P3000™, a transfection enhancer reagent provided in lipofectamine 3000™ kit, was also added into diluted plasmid tube to a final concentration of 2 µL/µg DNA. The diluted DNA was mixed with the diluted lipofectamine 3000™ at 1:1 ratio (v/v) and incubated for 15 min at room temperature. The plasmid-lipofectamine 3000™ complex was then added onto cells.” (Line 136-146),

Purification of DNA plasmids procedure was mentioned 

 “The recombinant plasmids were propagated in E. coli, DH5-alpha (Invitrogen, Carlsbad, CA, USA) and purified by Qiagen endotoxin-free giga plasmid kit (Hilden, Germany) following the manufacturer’s protocol.”(Line 130-133)

• In house preparation of S2 should be further explained.

Response: The procedure of S2 production and purification is described 

“The S2 protein was prepared by induction of the plasmid pET15b encoding S2 region fused with 6HIS at N terminal. In brief, transformed E. coli BL21(DE3) were grown at 37°C in LB (Luria-Bertani) medium supplemented with 100 µg/mL ampicillin. When the OD600 of the cell culture reached 0.6, isopropyl β-D-1-thiogalactopyranoside, IPTG (final concentration 0.1 mM) was added to induce protein expression and cells were grown at 25°C for a further 3 h. Cells were harvest by centrifugation at 8,000 g for 10 min at 4°C. For S2 purification, the cell pellet was resuspended in PBS supplemented with DNase (10 µg/mL) and RNase (10 µg/mL). The cells were lysed by sonication on ice for 15 min. The lysed cells were then centrifuged at 40,000 rpm for 1 h, only supernatant was collected. S2 protein was purified by affinity chromatography using HisTrap column (GE Healthcare, Sweden). The concentration of purified protein was measured by using Pierce™ BCA protein assay kit (Thermo Fisher Scientific, MA, USA).”(Line 165-176)

• Fig. 2: size bars are poorly visible and

Response: The bar size was improved

• Fig. 3: It is not clear which assay was used for IgG1xIgG2, please explain.

Response: IgG1 and IgG2a were analyzed by ELISA using secondary antibodies which specific to IgG1 and IgG2a subclasses. The method is mention in Line 182-185.

“The plates were then washed and incubated with 1:5000 dilutions of each horseradish peroxidase (HRP)-conjugated secondary Abs including rabbit anti-mouse IgG (KPL, USA), rabbit anti-mouse IgG1 or rabbit anti-mouse IgG2a (both Abs were from BioLegend, USA) for additional 1 h at 37°C.”

---

## [Decision Letter · Decision Letter 1]

11 Feb 2021

PONE-D-20-38402R1

DNA vaccine candidate encoding SARS-CoV-2 spike proteins elicited potent humoral and Th1 cell-mediated immune responses in mice

PLOS ONE

Dear Dr. Ketloy,

Thank you for submitting your manuscript to PLOS ONE. After careful consideration, we feel that it has merit but does not fully meet PLOS ONE’s publication criteria as it currently stands. Therefore, we invite you to submit a revised version of the manuscript that addresses the points raised during the review process:

1) Please, provide the gene sequence as supplemental information as requested by reviewer #2

We look forward to receiving your revised manuscript.

Kind regards,

Paulo Lee Ho, Ph.D.

Academic Editor

PLOS ONE

Reviewers' comments:

Reviewer's Responses to Questions

**Comments to the Author**

1. If the authors have adequately addressed your comments raised in a previous round of review and you feel that this manuscript is now acceptable for publication, you may indicate that here to bypass the “Comments to the Author” section, enter your conflict of interest statement in the “Confidential to Editor” section, and submit your "Accept" recommendation.

Reviewer #1: All comments have been addressed

Reviewer #2: All comments have been addressed

2. Is the manuscript technically sound, and do the data support the conclusions?

Reviewer #1: Yes

Reviewer #2: Yes

3. Has the statistical analysis been performed appropriately and rigorously? 

Reviewer #1: Yes

Reviewer #2: Yes

4. Have the authors made all data underlying the findings in their manuscript fully available?

Reviewer #1: Yes

Reviewer #2: No

5. Is the manuscript presented in an intelligible fashion and written in standard English?

Reviewer #1: Yes

Reviewer #2: Yes

6. Review Comments to the Author

Reviewer #1: After read, the last version, I concluded that all the comments were addressed, the manuscript is ready for publication.

Reviewer #2: Most of the required details were provided. However, the authors should provide a DNA sequence of the construct used for expression.

7. PLOS authors have the option to publish the peer review history of their article (what does this mean?). If published, this will include your full peer review and any attached files.

Reviewer #1: No

Reviewer #2: **Yes: **Sergio Schenkman

---

## [Author Response · Author response to Decision Letter 1]

11 Feb 2021

As the reviewer suggested the authors to provide a DNA sequence of the construct used for expression, the nucleotide sequences (humanized codons) of Spike (S), S1 and S2 are attached as supplementary data 1 (line 128).

---

## [Editor Report · Decision Letter 2]

18 Feb 2021

DNA vaccine candidate encoding SARS-CoV-2 spike proteins elicited potent humoral and Th1 cell-mediated immune responses in mice

PONE-D-20-38402R2

Dear Dr. Ketloy,

We’re pleased to inform you that your manuscript has been judged scientifically suitable for publication and will be formally accepted for publication once it meets all outstanding technical requirements.

Kind regards,

Paulo Lee Ho, Ph.D.

Academic Editor

PLOS ONE
---

## [Editor Report · Acceptance letter]

10 Mar 2021

PONE-D-20-38402R2 

DNA vaccine candidate encoding SARS-CoV-2 spike proteins elicited potent humoral and Th1 cell-mediated immune responses in mice 

Dear Dr. Ketloy:

I'm pleased to inform you that your manuscript has been deemed suitable for publication in PLOS ONE. Congratulations! Your manuscript is now with our production department. 

Kind regards, 

on behalf of

Dr. Paulo Lee Ho 

Academic Editor

PLOS ONE